# MOFI: Learning Image Representations from Noisy Entity Annotated Images

**Wentao Wu**[∗], **Aleksei Timofeev**[∗], **Chen Chen, Bowen Zhang, Kun Duan, Shuangning Liu,**
**Yantao Zheng, Jonathon Shlens**[†]**, Xianzhi Du, Yinfei Yang**
Apple AI/ML
{wentao_wu,a_timofeev,xianzhi,yinfeiy}@apple.com

## Abstract

We present **MOFI**, **M**anifold **OF I**mages, a new vision foundation model designed to learn image representations from noisy entity annotated images. MOFI differs from previous work in two key aspects: ($i$) pre-training data, and ($ii$) training recipe. Regarding data, we introduce a new approach to automatically assign entity labels to images from noisy image-text pairs. Our approach involves employing a named entity recognition model to extract entities from the alt-text, and then using a CLIP model to select the correct entities as labels of the paired image. It's a simple, cost-effective method that can scale to handle billions of web-mined image-text pairs. Through this method, we have created **Image-to-Entities (I2E)**, a new dataset with 1 billion images and 2 million distinct entities, covering rich visual concepts in the wild. Building upon the I2E dataset, we study different training recipes like supervised pre-training, contrastive pre-training, and multi-task learning. For constrastive pre-training, we treat entity names as free-form text, and further enrich them with entity descriptions. Experiments show that supervised pre-training with large-scale fine-grained entity labels is highly effective for image retrieval tasks, and multi-task training further improves the performance. The final MOFI model achieves 86.66% mAP on the challenging GPR1200 dataset, surpassing the previous state-of-the-art performance of 72.19% from OpenAI's CLIP model. Further experiments on zero-shot and linear probe image classification also show that MOFI outperforms a CLIP model trained on the original image-text data, demonstrating the effectiveness of the I2E dataset in learning strong image representations. We release our code and model weights at https://github.com/apple/ml-mofi.

## 1 Introduction

Over the past decade, the research community has devoted significant efforts to studying the acquisition of high-quality, general-purpose image representations (Donahue et al., 2014; Sun et al., 2017; Juan et al., 2019; Dosovitskiy et al., 2021). An effective image representation can yield impressive results on downstream tasks such as image classification and image retrieval across various domains, without requiring further customization.

Arguably, the most classical image representation learning method is based on *supervised* image classification (Deng et al., 2009; Sun et al., 2017), often using datasets like ImageNet and ImageNet21K (Deng et al., 2009). However, these datasets usually require expensive and difficult human labeling of precise class labels, which makes them less scalable. While some industrial labs have created large classification datasets using semi-automatic pipelines like JFT (Sun et al., 2017) or private data sources like IG hashtags (Singh et al., 2022b), how to further scale the datasets remains a challenge for the research community. Another prevailing approach to learn general image representations is leveraging the weakly supervised signals from text, which is easier to acquire and scale. For instance, state-of-the-art models like CLIP (Radford et al., 2021) and ALIGN (Jia et al., 2021) learn from billions of web-mined image-text pairs using a contrastive learning objective. Such pre-trained models can achieve strong zero-shot generalization results on various downstream tasks including image-text retrieval and image classification.

---

[∗]Equal contribution. [†]Work done while at Apple.

| Model | GPR1200 mAP@all (%) | ImageNet-ZS Acc@1 (%) |
|---|---|---|
| CLIP-L/14$_{OpenAI}$ | 72.19 | 75.27 |
| MOFI-L/14 | 86.15 | 77.17 |
| MOFI-H/14 | 86.66 | 78.46 |

(a) Comparison of MOFI and CLIP (Radford et al., 2021) on GPR1200 image retrieval and ImageNet zero-shot classification tasks.

| Dataset | # Images | # Classes |
|---|---|---|
| ImageNet-1K | 1.2M | 1K |
| ImageNet-21K | 14M | 21K |
| JFT-300M | 300M | 18K |
| JFT-3B | 3B | 30K |
| IG-3.6B | 3.6B | 27K |
| I2E (Ours) | 1.1B | 2M |

(b) I2E and existing large image classification datasets.

Figure 1: MOFI is trained on the new Image-to-Entities (I2E) dataset, which has 66x more classes than the previous datasets, and achieves significantly better performance on the image retrieval tasks.

Despite the great success of CLIP and ALIGN, they have not been able to explore the classification objective due to the typically varying associated text for each image. However, recent studies have demonstrated that incorporating supervised data (Pham et al., 2023; Zhai et al., 2022b) or improving data quality (Gadre et al., 2023; Cao et al., 2023) can enhance the performance of contrastive models. With these motivations in mind, we ($i$) investigate the potential of extracting entity labels from noisy image-text pairs, and ($ii$) training models to learn from these extracted labels.

First, we present a simple approach to automatically label images with entities *at scale*. Our method leverages existing noisy image-text datasets used for CLIP training. Given an image-text pair, a named entity recognition model is first applied to extract entities from the text. Each extracted entity is then paired with the original image and scored by a pre-trained CLIP model, and those image-entity pairs with low CLIP scores are filtered out. The constructed dataset, termed ***Image-to-Entities (I2E)***, consists of 1.1B images with 2M unique entities. To our best knowledge, I2E has the largest number of class labels documented thus far, 66 times more than JFT-3B (Zhai et al., 2022a) and IG-3.6B datasets (Singh et al., 2022a) (Table 1b). Compared with original noisy image-text data, entity labels contain more *structured* knowledge, which can potentially lead to better pre-trained models.

We study different training recipes to learn from the I2E dataset, including supervised pre-training, contrastive pre-training (CLIP), and multi-task learning. For the latter two, we treat entity names as free-form text and add entity descriptions of the entity to the text. The models are first evaluated on the image retrieval[1] benchmark GPR1200 (Schall et al., 2021), and a modified image retrieval task from ImageNet. Experimental results show that the CLIP model trained on the I2E data significantly outperforms the model trained on the original image-text data. Changing the training objective to supervised classification boosts performance even more. This shows both the I2E data and the classification objective are very effective for image retrieval tasks. The multi-task model reaches a new state-of-the-art of 86.15% mAP@all on GPR1200, beating the previous record of 72.19% from OpenAI's CLIP model (Table 1a). We also observe a significant performance gain 2.74% on the ImageNet image retrieval task. Given its strong performance on image retrieval, we name the multi-task model ***MOFI***, standing for **M**anifold **OF** **I**mages.

We further evaluate the models on standard ImageNet (Deng et al., 2009) and VTAB (Zhai et al., 2020) image classification tasks.[2] MOFI trained on the I2E data performs strongly compared to the CLIP model trained on the original image-text data. Specifically, for the ViT B/16 architecture (Dosovitskiy et al., 2021), MOFI achieves 72.99% zero-shot and 81.32% linear probe top-1 accuracy on ImageNet, significantly outperforming CLIP by 4.27% and 1.78%, respectively (evidenced later in Table 3 and 4 ). It also achieves the best linear probe and competitive zero-shot performance on VTAB tasks, with significant improvements on fine-grained recognition tasks like OxPet and OxFlowers.

Our contributions are summarized as follows.

- In terms of ***data***, we introduce ***Image-to-Entities (I2E)***, a new large-scale dataset with 1 billion images and 2 million distinct entities, covering rich visual concepts in the wild.
- In terms of ***model training***, we study various learning approaches from the constructed I2E dataset, including supervised pre-training, contrastive pre-training, and multi-task learning.

---

[1]**Image retrieval** by default means retrieving images that are similar to a query image, which is different from the image-text retrieval tasks used to evaluate CLIP. Image based retrieval has wide industrial use cases.

[2]For VTAB evaluation, we employ the 8 tasks from nature and specialized categories.

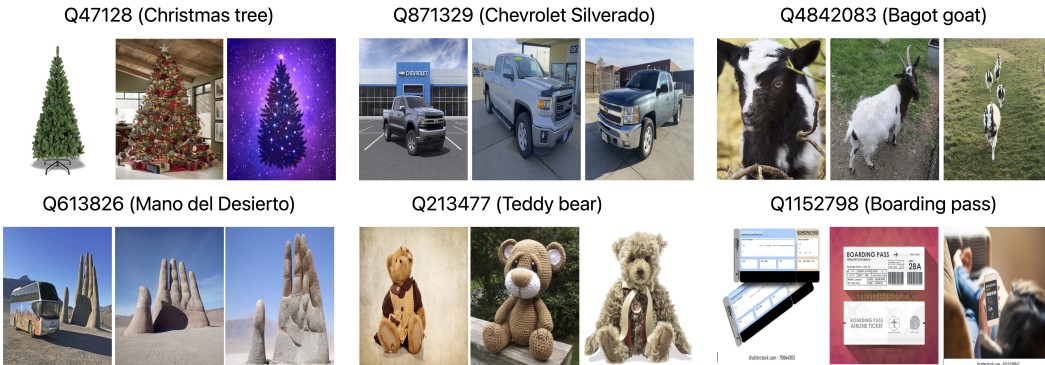

Figure 2: **Examples of the I2E dataset.** Each caption is formatted as Entity_id (Entity_name).[3]

- In terms of ***performance***, we advance the image retrieval SoTA on the GPR1200 dataset by a significant margin, and show that learning from the I2E data leads to strong image representations, with improved zero-shot performance on ImageNet and VTAB benchmarks.

## 2 NOISY IMAGE-TO-ENTITIES DATA AT WEB SCALE

In this section, we introduce our new approach to construct the Image-to-Entities (I2E) dataset.

### 2.1 METHOD

Image-text datasets are easily accessible through large-scale web crawled corpus. This has been widely used for training vision foundation models such as CLIP (Radford et al., 2021) and ALIGN (Jia et al., 2021). The texts are typically gathered from webpages, *e.g.*, image alt-text and page titles, and are often unstructured and in free-form. We build the *structured* I2E dataset on top of this. At a high level, the steps are: *1. Construct image-text dataset from crawled web corpus; 2. Extract entities from text; 3. Entity filtering; 4. Sample the resulting dataset.* Next, we describe each step in detail.

**Constructing image-text dataset.** Following CLIP (Radford et al., 2021) and ALIGN (Jia et al., 2021), we remove pornographic images based on a NSFW model and images whose shorter dimension is less than 200 pixels from crawled web corpus. The images are further deduplicated based on image hash. For each selected image, we select both alt-text and page title as text candidates. The text candidates are further filtered based on text length, full text frequency, unigram and bigram frequency, *etc.* Similar to Schuhmann et al. (2021), we use OpenAI CLIP-B/16 model to compute image and text embeddings, and compute the cosine similarity between them and remove image-text pairs below a threshold of 0.24. From our crawled web corpus, we constructed a large-scale image-text dataset which contains 8.4B image-text pairs. As one image may have multiple texts, it contains 5.5B images.

**Extracting entities from text.** In order to extract entities from text which can be used as labels for model pre-training, we not only need to locate the named entities in the text (via named entity recognition), but also need to assign a unique identifier to each entity (via named entity linking).

To do this, we first find all possible entity candidate based on all $n$-grams and compute its probability based on its popularity from wikipedia. As entities may have ambiguity if purely based on text, *e.g.*, Apple company *vs.* Apple fruit, we mitigate this problem by using other entities in the same text. Specifically, we first pre-compute an embedding for every entity as described later. For each candidate, we update its probability in an iterative way until convergence. In each iteration, we first compute the embedding for the full text by combining all entity candidates' embedding based on its probability, then update the probability for each candidate based on its distance to the full text embedding. In the end, we select the best candidate with the highest probability.

Similar to Nickel & Kiela (2017), the entity embeddings are hyperbolic graph embedding trained on the Wikipedia link graph and click stream data. The loss is to minimize the distance between two

---

[3]Entities can be found through `https://www.wikidata.org/wiki/[Entity_id]`.

Figure 3: **Illustration of different approaches explored in this paper** to learn image representations from I2E dataset. Supervised pre-training treats entities as labels, contrastive pre-training uses entity names and descriptions as free-form text, and multi-task pre-training combines the two.

entities occurring together on Wikipedia. For simplicity, we utilize the Wikidata knowledge base and its associated identifiers for items. This approach has reasonable precision and recall, and can be run efficiently at large scale. We are able to annotate 8.4B texts in less than 10 hours with 256 machines, each has 30GB memory.

**Entity filtering.** The entities extracted from text may not be always related to the corresponding image: We may not be able to disambiguate entities when there is not enough context, especially when most texts are short. Even if the entities are correctly extracted from text, it may not be always relevant to the image. Take an image of Honda Civic with text "Used Honda Civic for Sale in Los Angeles, CA" as an example. The text is considered as relevant based on the image-text embedding, and we can extract two entities from the text, Honda Civic and Los Angeles, both are correct from the text perspective, but clearly the latter is not relevant to the image.

In order to reduce the noise in the dataset, we use CLIP model to compute the CLIP embedding for the text representation of entities, and filter out the entities which have lower cosine similarity with the image embedding. To further leverage *external knowledge* for enhanced performance (Shen et al., 2022), the entity names are further enriched with entity descriptions, *e.g.*, from Wikidata. Therefore, the text representation of an entity is formed as entity name, entity description.

**Sampling the resulting dataset.** After filtering, the dataset contains 1.24B images with around 3.1M entities. Although it contains a large number of fine-grained entities, the distribution of number of images per entity is highly skewed to the popular entities (see Table 1). To ease pre-training, we removed all the images associated with entities which have less than 5 images.

## 2.2 STATISTICS

The constructed dataset contains 1.1B images with 2.1M entities. To our knowledge, this is one of the largest weakly labelled datasets available, with the largest number of entities, approximately 66 times more than JFT-3B (Zhai et al., 2022a) and IG-3.6B (Singh et al., 2022a), and around 100 time more than ImageNet-21k (Deng et al., 2009) and JFT-300M (Sun et al., 2017).

To differentiate the constructed dataset with the original image-text dataset, we call the new dataset **I2E** (Image-to-Entities), and the original dataset **I2T** (Image-to-Text) in the rest of the paper. The distribution of number of images per entity is shown in Table 1. It is highly skewed to popular entities. We observe that the apparel entities are the most popular ones, *e.g.*, Q131151(T-shirt) and Q83363(jeans) have 4.7M and 2.3M images, respectively. Examples of the I2E dataset are provided in Figure 2.

Table 1: **Distribution of number of images per entity.** Note that entities in the $[0, 5)$ range are removed in our final dataset.

| Range | # Entities |
|---|---|
| $[0, 5)$ | $955,651$ |
| $[5, 10)$ | $418,895$ |
| $[10, 100)$ | $1,127,198$ |
| $[100, 1000)$ | $506,753$ |
| $[1000, 10000)$ | $117,802$ |
| $[10000, \inf)$ | $18,768$ |

## 3 LEARNING FROM IMAGE-TO-ENTITIES DATA

In this section, we describe how the I2E dataset is used for learning general-purpose image representations. We explore three types of learning approaches, as illustrated in Figure 3.

### 3.1 SUPERVISED PRE-TRAINING

The image classification task is one of the simplest, but very effective methods of using labeled data for learning image representations. In our context, each entity is considered to be a separate label, and the classification task is to predict which labels correspond to the given image.

There are multiple choices of loss functions which can be used to promote embedding properties such as separability *etc*. In experiments, we use the large margin cosine loss in Wang et al. (2018), as it was shown to be simple and effective, on par with other more complicated methods (Musgrave et al., 2020). Since the I2E dataset has an immense number of entities (over 2 million), predicting all the entities in each batch is computationally costly. Similar to sampled softmax (Tensorflow_Authors), a fixed number of entities is used for each batch - entities of the in-batch images plus entities randomly sampled to be used as additional negatives. The exact number of entities used for each batch was selected based on a trade-off between performance and quality[4]. Formally,

$$\mathcal{L}_{\text{class}} = \sum_{k \in B} \log \frac{e^{(\langle e_k^{im}, w_{c_k} \rangle - m)/t}}{e^{(\langle e_k^{im}, w_{c_k} \rangle - m)/t} + \sum_{c \in C_k} e^{\langle e_k^{im}, w_c \rangle / t}} \,, \tag{1}$$

where $e_k^{im}$ denotes the image embedding for the $k$-th sample, with $c_k$ the corresponding class label. $C_k = C' \setminus \{c_k\}$, $C' = C_{batch} \cup C_{random}$, $C_{batch} = \{c_i | i \in B\}$, $C_{random} = \{c | c \sim \mathcal{U}[C \setminus C_{batch}]\}$, and $\mathcal{U}[C]$ denotes uniform distribution over classes. Size of $C_{random}$ is selected to achieve $|C'| = N$. $m = 0.15$ and $t = \frac{1}{32}$ are margin and temperature correspondingly. $w_c$ is the embedding of class $c$.

### 3.2 CONTRASTIVE PRE-TRAINING

Contrastive learning of image and text correspondence is another popular way of weakly-supervised pre-training of image representations (Radford et al., 2021; Jia et al., 2021). Given a set of image-text pairs $(I_k, T_k)$, the goal is to learn embedding $e_k^{im} = f_{im}(I_k)$ and $e_k^{txt} = f_{txt}(T_k)$ such that the similarity $\langle e_k^{im}, e_k^{txt} \rangle$ is larger than $\langle e_k^{im}, e_j^{txt} \rangle$ and $\langle e_j^{im}, e_k^{txt} \rangle$ for $j \neq k$. Thus, the following cross-entropy loss is used for model training for a batch $B = \{I_k, T_k\}_{k=1}^K$. Specifically,

$$\mathcal{L}_{\text{contrast}}^{im} = \sum_{k \in B} \log \frac{e^{\langle e_k^{im}, e_k^{txt} \rangle / \tau}}{\sum_{j \in B} e^{\langle e_k^{im}, e_j^{txt} \rangle / \tau}} \,, \quad \mathcal{L}_{\text{contrast}}^{txt} = \sum_{k \in B} \log \frac{e^{\langle e_k^{im}, e_k^{txt} \rangle / \tau}}{\sum_{j \in B} e^{\langle e_j^{im}, e_k^{txt} \rangle / \tau}} \,, \tag{2}$$

$$\mathcal{L}_{\text{contrast}} = L_{\text{contrast}}^{im} + L_{\text{contrast}}^{txt} \,, \tag{3}$$

where $\tau$ is a temperature which is also learned during model training.

### 3.3 MOFI: MULTI-TASK PRE-TRAINING

In the final setup, we combine the entity-based image classification loss with the image-text-based contrastive loss to learn a universal image representation. In this setup, a text embedding is produced, which is compatible with the learned image representation, and can be used for zero-shot image classification, *etc*. Since entities are extracted directly from text, each training example already consists of a triple of aligned image, text(s) and entities. Thus, it is straightforward to train the model with the above losses together. Specifically,

$$\mathcal{L}_{\text{combined}} = \lambda \mathcal{L}_{\text{class}} + (1 - \lambda) \mathcal{L}_{\text{contrast}} \,, \tag{4}$$

where $\lambda$ is a hyper-parameter to balance the two loss terms. For simplicity, we set $\lambda$ to 0.5 in all experiments. Note that compared with the plain alt-text used for model training, we explore the use of entity name and entity descriptions as *external knowledge* for better performance.[5]

---

[4]More details and ablation study can be found in Appendix C.

[5]Similar ideas have also been explored in K-Lite (Shen et al., 2022); however, the scale in K-Lite is much smaller (28M images compared with 1B images used in our training), while ours serves as the first evidence to show external knowledge can be useful at scale.

Table 2: **Results on image retrieval.** mAP@all is reported on GPR1200, and kNN classification accuracy, Recall@1, and Recall@5 are reported on ImageNet-1K. I2T is the original noisy image-text dataset, based on which we construct our I2E dataset. Image retrieval refers to *image-to-image* retrieval. **Bold** and underline are used to indicate the highest and second highest values in each bucket, respectively.

| Row | Model | Data | GPR1200 | | | | | | | ImageNet-1K | | |
|-----|-------|------|-----|-------|-------|------|------|--------|-------|------|-------|-------|
| | | | All | Land. | Faces | iNat | INST | Sketch | SOP | Acc$_{kNN}$ | Acc@1 | Acc@5 |
| 1 | ViT-L (Schall et al., 2021) | IN21k | 63.2 | 84.9 | 25.3 | 45.0 | 60.4 | 74.8 | 88.8 | - | - | - |
| 2 | Swin-B (Schall et al., 2021) | IN21k+GLv2 | 66.2 | 91.7 | 29.5 | 50.3 | 74.85 | 59.2 | 91.4 | - | - | - |
| 3 | CLIP-B/16$_{OpenAI}$ (Radford et al., 2021) | Web-400M | 67.33 | 87.52 | 73.01 | 32.59 | 73.72 | 51.29 | 85.87 | 74.10 | 61.82 | 82.02 |
| 4 | CLIP-B/16$_{Ours}$ | I2T | 65.94 | 87.43 | 60.64 | 31.66 | 74.73 | 49.05 | **92.17** | 74.41 | 62.50 | 82.48 |
| 5 | DINOv2-B/14 (Oquab et al., 2023) | LVM-142M | 57.44 | 91.84 | 8.89 | 35.50 | 73.74 | 51.40 | 83.26 | **82.1** | **76.6** | **89.0** |
| 6 | CLIP-B/16 | I2E | 76.14 | 91.74 | 84.32 | 41.78 | 80.16 | 68.39 | 90.41 | 78.55 | 69.70 | 85.82 |
| 7 | MOFI$_{SupOnly}$-B/16 | I2E | **83.37** | **95.55** | 97.23 | 53.85 | **85.99** | 80.69 | 86.92 | 79.66 | 71.68 | 86.68 |
| 8 | MOFI-B/16 | I2E | 83.33 | 95.45 | **97.24** | **54.45** | 84.16 | **81.02** | 87.67 | 80.27 | 73.34 | 87.12 |
| 9 | CLIP-L/14$_{OpenAI}$ (Radford et al., 2021) | Web-400M | 72.19 | 89.88 | 79.96 | 36.85 | 78.52 | 58.67 | **89.24** | 79.77 | 70.14 | 87.32 |
| 10 | DINOv2-L/14 (Oquab et al., 2023) | LVM-142M | 61.23 | 92.58 | 8.99 | 36.82 | 82.82 | 59.75 | 86.41 | **83.5** | **78.04** | **89.72** |
| 11 | MOFI-L/14 | I2E | **86.15** | **96.71** | **98.44** | **63.41** | **84.97** | **85.5** | 87.89 | 82.51 | 76.78 | 88.98 |
| 12 | MOFI-H/14 | I2E | 86.67 | 96.96 | 98.62 | 65.52 | 84.0 | 86.82 | 88.05 | 82.73 | 77.60 | 89.34 |

We name the multi-task learned model **MOFI**, standing for **M**anifold **OF** **I**mages, and the supervised learned model **MOFI$_{SupOnly}$**. We still use **CLIP** to refer to the constrastive model learned from I2E.

## 4 EXPERIMENTS

We perform comprehensive experiments to evaluate the performance of MOFI models learned from the I2E dataset. We compare with SoTA models under image retrieval (Section 4.2) and image classification tasks (both linear probe and zero-shot evaluation as detailed in Section 4.3).

### 4.1 SETUP

We employ a transformer (Vaswani et al., 2017) architecture as the backbone in all experiments. The default MOFI-B/16 model adopts the CLIP-B/16 (Radford et al., 2021) configuration for the image encoder, which consists of a 12-layer transformer with 12 attention heads and 768-dimension hidden feature, which is projected to 512 dimension as the final image embedding. When training with contrastive objectives, we also employ the text encoder configuration from CLIP-B/16, which is 12 transformer layers with 8 attention heads and 512 feature dimension. The input text is tokenized by the OPT tokenizer (Zhang et al., 2022) with a vocabulary size of 50,265. The maximum input sequence length is set to 76.

For MOFI-L/14, the image encoder is a 24-layer transformer with 16 heads and 1024-dimension hidden feature, which is projected to 512 dimension as output; the text encoder is a 12-layer transformer with 12 heads and 768 dimension feature. For MOFI-H/14 model, the image encoder is a 32-layer transformer with 16 heads and 1280-dimension hidden feature, which is projected to 1024 dimension as output; the text encoder is a 24-layer transformer with 16 heads and 1024 dimension feature.

All models are trained with 224x224 input image size using the AdamW optimizer (Loshchilov & Hutter, 2017) with weight decay 0.1 and learning rate 0.0008, except that MOFI-L/14 uses a learning rate of 0.0006. The learning rate is first warmed up to 10,000 steps, and cosine decay is applied until the last training step. Due to the computation limit, we train the CLIP models for 600k steps with global batch size 32,768, and train the other models for 1.2M steps with global batch size 16,384, so all the models have seen the same number of training examples. The number of entities $N$ used in classification for each batch is set to 512k. All experiments are conducted using AXlearn[6].

### 4.2 RESULTS ON IMAGE RETRIEVAL

We first evaluate the models on image retrieval tasks on GPR1200 (Schall et al., 2021) and ImageNet-1K (Russakovsky et al., 2015)[7]. GPR1200 is a general-purpose content-based image retrieval benchmark, which consists of subsets from six diverse domains. In total, there are 1200 categories,

---

[6] https://github.com/apple/axlearn

[7] We report additional image retrieval tasks $\mathcal{R}$Oxford and $\mathcal{R}$Paris (Radenovic et al., 2018) in Appendix A.

Table 3: **Zero-shot classification accuracy.** The top-1 accuracies (%) across ImageNet and 9 VTAB tasks (6 tasks from natural and 3 tasks from specialized sets) are reported.

| Model | ImageNet | VTAB | | | | | | | | | |
|---|---|---|---|---|---|---|---|---|---|---|---|
| | | Caltech101 | CIFAR100 | SVHN | DTD | OxPet | OxFlowers | Eurosat | RESISC45 | Camelyon | Avg |
| CLIP-B/16$_{OpenAI}$ | 68.38 | 81.59 | 65.66 | 52.09 | 43.24 | 88.03 | 71.72 | 51.50 | 64.68 | **56.89** | 63.94 |
| CLIP-B/16$_{ours-I2T}$ | 68.72 | 83.91 | 66.88 | 51.69 | **59.20** | 82.31 | 68.30 | 48.48 | 65.27 | 55.23 | 64.59 |
| CLIP-B/16$_{ours-I2E}$ | 72.94 | 84.38 | **66.89** | 45.98 | 51.59 | 86.89 | 78.08 | **53.50** | **65.65** | 55.69 | **65.40** |
| MOFI-B/16 | **72.99** | **85.14** | 64.44 | **56.24** | 48.94 | **90.02** | **78.82** | 46.63 | 61.20 | 52.60 | 64.89 |
| CLIP-L/14$_{OpenAI}$ | 75.31 | 83.69 | **76.03** | **55.68** | 51.75 | 93.02 | 77.80 | **60.92** | **71.27** | **58.17** | **69.81** |
| MOFI-L/14 | **77.18** | **86.40** | 73.56 | 53.51 | **55.32** | **94.93** | **83.17** | 51.35 | 64.06 | 50.51 | 68.09 |
| MOFI-H/14 | 78.46 | 85.78 | 75.26 | 52.71 | 60.00 | 96.10 | 85.20 | 59.28 | 69.71 | 50.82 | 70.54 |

Table 4: **Linear probe classification accuracy.** The top-1 accuracies (%) across ImageNet and 9 VTAB tasks (6 tasks from natural and 3 tasks from specialized sets) are reported.

| Model | ImageNet | VTAB | | | | | | | | | |
|---|---|---|---|---|---|---|---|---|---|---|---|
| | | Caltech101 | CIFAR100 | SVHN | DTD | OxPet | OxFlowers | Eurosat | RESISC45 | Camelyon | Avg |
| CLIP-B/16$_{OpenAI}$ | 80.2 | **94.7** | **83.1** | – | 79.2 | 93.1 | 98.1 | 92.7 | 92.7 | – | – |
| CLIP-B/16$_{ours-I2T}$ | 79.54 | 93.36 | 77.37 | 70.35 | **83.46** | 91.69 | 98.29 | 95.81 | 93.94 | 82.87 | 87.46 |
| CLIP-B/16$_{ours-I2E}$ | 81.03 | 92.21 | 77.94 | 68.36 | 81.91 | 94.28 | 99.43 | 96.74 | 93.48 | **85.15** | 87.72 |
| MOFI-B/16$_{SupOnly}$ | 80.53 | 88.35 | 78.94 | 70.74 | 80.37 | 95.26 | 99.54 | **96.78** | **94.52** | 85.00 | 87.72 |
| MOFI-B/16 | **81.32** | 90.47 | 77.87 | **70.87** | 82.66 | **95.56** | **99.58** | 96.50 | 94.24 | 85.08 | **88.09** |

and each category has 10 images. Follows its original paper, images are not split as query and index sets for evaluation, we retrieve the nearest neighbor for every image and use the rest as index set. We report the full mean Average Precision (mAP) @all for the entire dataset and each domain. For ImageNet, we modify its validation set to use as an image retrieval evaluation set. We randomly select one image from each category as query set, and use the rest as index set. For each retrieval result, we consider it as positive if it has the same category as the query; otherwise, we consider it as negative. We report the top-1 and top-5 accuracy for this dataset. For kNN metric, we follow Wu et al. (2018) and train a kNN classifier on the training set of ImageNet-1K (Russakovsky et al., 2015). The best accuracy on the validation set over a hyper-parameters sweep was reported.

Results are summarized in Table 2. We compare MOFI with existing supervised models and different versions of CLIP models, *i.e.*, CLIP model from OpenAI trained on 400M dataset (Row 3), CLIP model trained on our internal 5.5B image-text dataset (Row 4) and CLIP model trained from our 1.1B I2E dataset (Row 6). On GPR1200, MOFI outperforms existing supervised model (+17.13, Row 8 *vs.* Row 2) and CLIP model (+16.0, Row 8 *vs.* Row 3) by a significant margin. It is interesting to see that on the Landmark domain, MOFI outperforms Swin-B (+3.75, Row 8 *vs.* Row 2), which is pre-trained on ImageNet21k and then finetuned on a clean domain specific dataset, *i.e.*, Google LandmarkV2 dataset. It is also worthwhile to mention that our model performs worse (-4.5, Row 8 *vs.* Row 4) on the SOP domain when compared to our CLIP model. We hypothesize this is due to that using our current data mining approach, it may be hard to extract fine-grained entities from product-related text. We leave further exploration on this as future work. On ImageNet1k, we observe +10.84 and +4.64 improvement on Acc@1 and Acc@5 compared to our CLIP model (Row 8 *vs.* Row 4).

For the sake of interest, we also compare our model with DINOv2 (Oquab et al., 2023), a recent model trained on a curated dataset (LDV-142M) consisting of 142 million images close to domains of a list of tasks. The DINOv2 model is reported with strong image retrieval performance. Table 2 shows that MOFI significantly outperforms DINOv2 on GPR1200, but is worse on ImageNet-1K. We believe this is due to that DINOv2's training data is curated for a target list of tasks. Apparently, ImageNet-1K is included in the targeted list and other domains are not included.

### 4.3 RESULTS ON IMAGE CLASSIFICATION

**Zero-shot evaluation.** MOFI can also be used for zero-shot image classification, as it is trained with a text encoder using contrastive loss. Table 3 summarizes results on ImagetNet and VTAB, employing the same prompt set from CLIP (Radford et al., 2021). MOFI achieves better or similar performance compared to CLIP on most datasets. For example, MOFI-B/16 improved over CLIP-B/16$_{ours-I2T}$ by 4.27 and 0.3 points on ImageNet and VTAB, respectively, despite the CLIP model is trained on 5x larger dataset. Notably, MOFI models excel in most natural tasks within VTAB, particularly in the

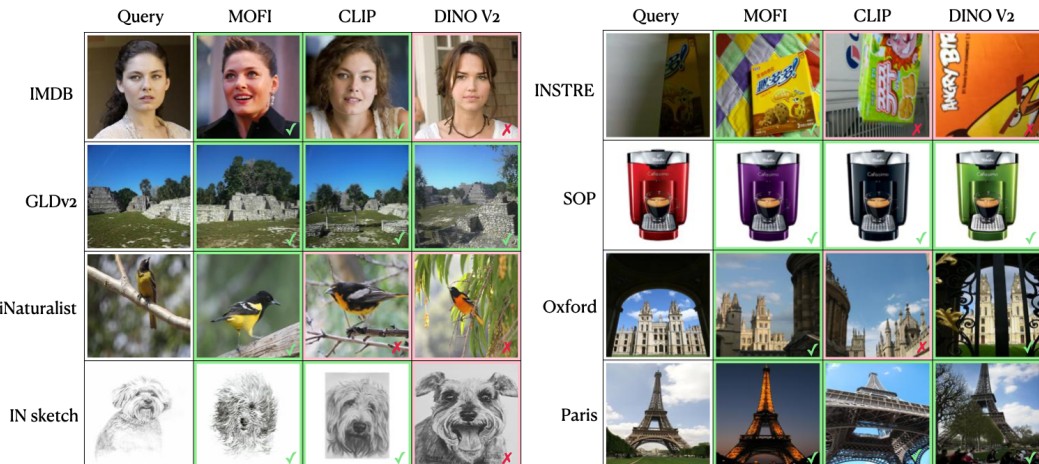

Figure 4: **Examples of top-1 retrieved images on GPR1200, $\mathcal{R}$Oxford and $\mathcal{R}$Paris** [7] **evaluation sets.** Green (✓) and red (✗) indicate positive or negative images, respectively.

domains of OxPet and OxFlowers, where precise object recognition is crucial. Nevertheless, these models struggle with specialized tasks, possibly because the images in those tasks present greater challenges in terms of entity description. Note that zero-shot classification also requires a strong text encoder. The primary goal of the MOFI model is to learn better image embeddings, and it takes a significant part of the computation budget.

**Linear probe.** We extract image features before the linear projection to the shared embedding space following CLIP (Radford et al., 2021), and train a logistic regression classifier on top of them for each dataset. We use the AdamW optimizer without weight decay. We conducted a hyper-parameter sweep on training epochs (*i.e.*, 10, 20, 40, 80, 160) and learning rates (*i.e.*, 1e-1, 1e-2, 1e-3, 1e-4) to identify the optimal configurations. Table 4 shows that MOFI performs the best, outperforming CLIP$_{\text{ours-I2T}}$ by an average of 1.78 and 0.63 on ImageNet and VTAB, respectively. When CLIP was trained using I2E data, or MOFI was trained with only classification objective, both models performed better than CLIP using I2T data, but worse than MOFI. This highlights the significance of a multi-task setup in achieving better performance.

## 4.4 ANALYSIS

In order to illustrate the difference between the learned embeddings, images from different sub-categories of GPR1200 (Schall et al., 2021) as well as images from $\mathcal{R}$Oxford and $\mathcal{R}$Paris (Radenovic et al., 2018) are used to retrieve the most similar images based on the embedding from different models. Results are shown in Figure 4. For each query image and each model, we show the most similar image from the corresponding index set. Images with the correct label (*i.e.*, the same label as the query) have a green frame and a check-mark (✓), others have a red frame and an x-mark(✗).

**Comparing the I2E and I2T datasets.** We quantitatively compare the I2E and I2T datasets for CLIP and MOFI model training on image retrieval and zero-shot classification tasks. Similar to the other experiments, we put the entity name together with its original text and sample the text with equal probability when using contrastive objectives. Results are reported in Table 5 for models trained on I2T, I2E, or combined. Even for the CLIP model, switching the dataset from I2T to I2E leads to a significant performance improvement on both image retrieval tasks. The performance on the classification tasks are close, with wins on ImageNet and loss on VTAB. The model trained on the combined I2E and I2T dataset is better than the model trained on I2T, but worse than the model trained on I2E. These results indicate that we can also use the entity mining process as a data cleaning and selection process to improve the original image-text dataset, which is aligned with the observation from Gadre et al. (2023); Cao et al. (2023).

## 5 RELATED WORK

**Supervised pre-training** on extensive human-labelled datasets, such as ImageNet and ImageNet21k (Deng et al., 2009), has emerged as a widely adopted approach to acquire transferable visual representations. This approach has greatly expedited progress in various computer vision

Table 5: **Comparision of CLIP and MOFI models on I2T and I2E data.** For image retrieval, we report mAP@all(%) on GPR1200-*All*, and Acc@1(%) on ImageNet-kNN classification. For zero-shot classification, we report Acc@1(%) on VTAB (averaged across 9 tasks) and ImageNet. The image encoder of all models use the ViT B/16 architecture. **Bold** and underline are used to indicate the highest and second highest values, respectively.

| Model | Data | Image Retrieval | | ZS Classification | |
|-------|------|-----------------|----------------|-------------------|-------|
|       |      | GPR1200 | $IN_{kNN}$ | VTAB | IN |
| CLIP | I2T | 65.94 | 74.41 | 64.59 | 68.72 |
| CLIP | I2E | 76.14 | 78.55 | 65.41 | 72.94 |
| CLIP | I2E + I2T | 73.98 | 77.93 | 64.80 | **73.09** |
| MOFI | I2E | 83.33 | **80.27** | 64.89 | 72.99 |
| MOFI | I2E + I2T | **83.41** | 80.13 | **65.58** | 72.99 |

tasks like image classification (Donahue et al., 2014; Sharif Razavian et al., 2014), object detection/segmentation (Girshick et al., 2014; Ren et al., 2015), and visual question answering (Chen et al., 2020b; Li et al., 2020). Nevertheless, the effectiveness of learned representations is frequently constrained by the scale and diversity of supervision within the pre-training dataset. For larger-scale pre-training, noisy labels can be also derived from noisy image-text pairs crawled from the web (Ghadiyaram et al., 2019; Mahajan et al., 2018), and certain industrial laboratories have successfully built comprehensive classification datasets by utilizing semi-automatic pipelines, such as JFT (Zhai et al., 2022a), or private data sources like Instagram hashtags (Singh et al., 2022a). (Hu et al., 2023) has a similar approach as ours to utilize Wikipedia entities as a unified label space. However, they combines 14 existing dataset and primarily focus on improving generalization from known to unknown labels with respect to a given text query.

**Contrastive pre-training** is another prominent approach for acquiring transferable image representations through text supervision. In particular, models such as CLIP (Radford et al., 2021), ALIGN (Jia et al., 2021) and Florence (Yuan et al., 2021) have showcased impressive zero-shot image classification and image-text retrieval capabilities by mapping images and text into a shared embedding space. In addition, CoCa (Yu et al., 2022) incorporates an additional image captioning objective alongside the contrastive loss, LiT (Zhai et al., 2022c) proposes the freezing of a pre-trained image encoder for contrastive training, and subsequent studies (Yao et al., 2021; Lee et al., 2022; Li et al., 2021; Mu et al., 2022; Wu et al., 2021; Yang et al., 2022; Weers et al., 2023) further enhance the contrastive training objective. Furthermore, research has been conducted on various aspects including non-contrastive learning (Zhou et al., 2023), the integration of external knowledge (Shen et al., 2022), masking (Li et al., 2023), sparse embedding (Chen et al., 2023), and more.

Our work distinguishes itself from previous studies in two key aspects. Firstly, in terms of data, we have curated a new dataset consisting of 1 billion images and 2 million distinct entities, making it the largest dataset of its kind. Secondly, regarding model training, we propose a new approach that combines supervised and contrastive pre-training, where supervised training treats entities as labels, while contrastive training utilizes entity names as text and augments them with entity descriptions.

Note that the community also explore image-only self-supervised learning methods (Chen et al., 2020a; He et al., 2020; Grill et al., 2020; Caron et al., 2021; Chen & He, 2021; Bao et al., 2021; He et al., 2022; Wei et al., 2022) We focus on learning from language supervision in this paper.

## 6 CONCLUSION

This paper introduces MOFI, a new vision foundation model derived from billion-scale noisy entity annotated images. To train MOFI, we first construct a large-scale dataset, called Image-to-Entities (I2E), consisting of 1 billion images and 2 million distinct entities derived from noisy image-text pairs. Subsequently, we explore three different training approaches, and show that supervised pre-training on a large number of entities significantly enhances the performance of image retrieval tasks, and multi-task learning achieves the best performance. Additionally, we demonstrate that MOFI yields robust image representations, as evidenced by enhanced zero-shot and linear probe performance on benchmarks such as ImageNet and VTAB, outperforming CLIP.

ACKNOWLEDGEMENTS

We would like to acknowledge Zhe Gan for his substantial contributions to this project, particularly for his insightful discussions regarding the experiments and thorough review of the paper. We also express our appreciation to Kushal Tayal, Madhav Sharan, and others for their foundational work in entity extraction from text. Furthermore, we acknowledge the invaluable feedback and suggestions from our teammates at Apple AIML. Special thanks go to Yang Zhao, Liangliang Cao, Xiangxin Zhu, and Vivek Rathod for their early review of the paper and for providing valuable insights and comments.

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

# Appendices

## A. ADDITIONAL IMAGE RETRIEVAL RESULTS

Table 6 reports results on two additional benchmarks, $\mathcal{R}$Oxford and $\mathcal{R}$Paris (Radenovic et al., 2018), which are also included in DINOv2's target task list. On these two tasks, MOFI outperforms CLIP models by a significant margin. When trained with GLDv2 (Weyand et al., 2020) data, which is in a similar domain as $\mathcal{R}$Oxford and $\mathcal{R}$Pairs, MOFI (f)[8] model achieves comparable performance to DINOv2 across all metrics. It is worth to note that DINOv2 uses smaller patch size and is also distilled from a larger g/14 model. Both models are still worse than specialized models that are designed for these tasks.

Table 6: **Additional image retrieval results** on $\mathcal{R}$Oxford and $\mathcal{R}$Paris. DINOv2 is trained with more in-domain data and distilled from a larger g/14 model.

| Model | $\mathcal{R}$**Oxford** | | $\mathcal{R}$**Paris** | |
|---|---|---|---|---|
| | Medium | Hard | Medium | Hard |
| *Generic models* | | | | |
| CLIP-B/16$_{OpenAI}$ | 36.84 | 12.26 | 67.34 | 44.70 |
| CLIP-B/16$_{Ours}$ | 38.32 | 11.16 | 69.41 | 47.27 |
| MOFI-B/16 | 67.33 | 36.93 | 86.08 | 71.75 |
| CLIP-L/14$_{OpenAI}$ | 44.82 | 18.22 | 68.59 | 47.44 |
| MOFI-L/14 | 70.97 | 41.20 | 85.55 | 71.97 |
| *Generic models with in-domain data* | | | | |
| MOFI-B/16 w/ GLDv2 | 69.24 | 39.08 | 88.35 | 76.15 |
| MOFI-B/16 w/ GLDv2 (f)[8] | 71.05 | 46.24 | 88.39 | 75.88 |
| DINOv2-B/14 (Oquab et al., 2023) | 72.90 | 49.50 | 90.30 | 78.50 |
| *Specialized models* | | | | |
| DOLG (Yang et al., 2021) | 80.5 | 58.8 | 89.8 | 77.7 |
| MSTFC (Mari et al., 2022) | 80.3 | 60.3 | 91.2 | 80.6 |

## B. VISUALIZATION OF THE MOFI IMAGE REPRESENTATION SPACE

We show the distribution of fine-grained classes in GPR1200 eval set in the feature space from MOFI in Figure 5. We represent each class with the average feature vector of its examples. We first reduce the feature dimension to 48, and then run t-SNE with a perplexity of 20, a learning rate of 50 for 300 iterations. The left figure shows that the six domains are grouped together. The *imdb* domain has the most concentrated distribution, as it primarily consists of face images, while the *stfproduct* domain has a more dispersed distribution, as it encompasses a wider range of diverse categories. The right figure shows the distribution of different product categories in *stfproduct* domain. The categories that are similar to each other are located closer together in the embedding space compared to those that are not similar, *e.g.*, *coffee maker* and *kettle* are more closely related than *fan* and *sofa*.

## C. ABLATION STUDY

In this section, we perform ablation study on the number of in-batch negatives for supervised pre-training, and the number of entities needed to achieve the best performance. Limited by time and computation, we train models with 300k steps, and the learning rate cosine decay to 0 at the 300k step. At last, we also compare the I2E and I2T data. We evaluate the model on two image retrieval tasks (GPR1200 and ImageNet), and two zero-shot classification tasks, including VTAB averaged across 9 tasks and ImageNet.

**In-batch negatives for supervised pre-training.** As described in Section 3.1, the supervised learning objective is required to predict from 2M entities, which is very computation costly and slow. To make training more efficient, a simple strategy that samples $N$ entities in each batch is used, which indicates

---

[8](f) indicates that the full image is resized to 224x224 regardless of its original aspect ratio, which is similar to DINOv2 setup. All other MOFI and CLIP models resize the shortest edge to 224 first and then perform a center crop.

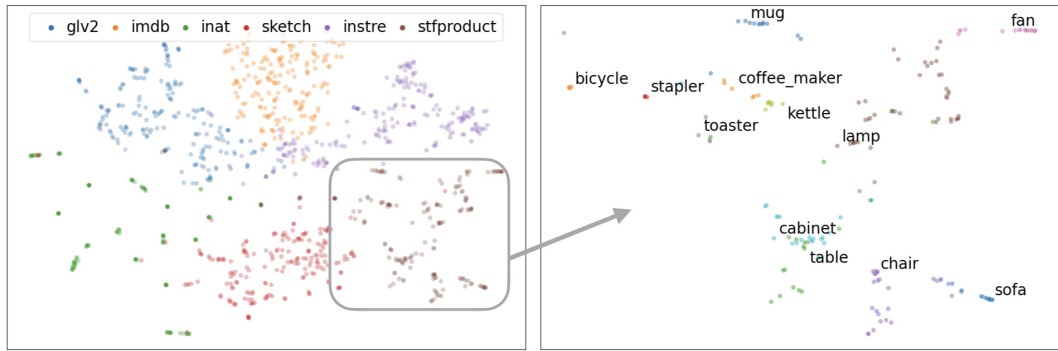

Figure 5: **t-SNE visualization of MOFI learned image representations on GPR1200 evaluation set**. The left figure shows the distribution of six domains in the feature space. The right figure shows the distribution in *stfproduct* domain.

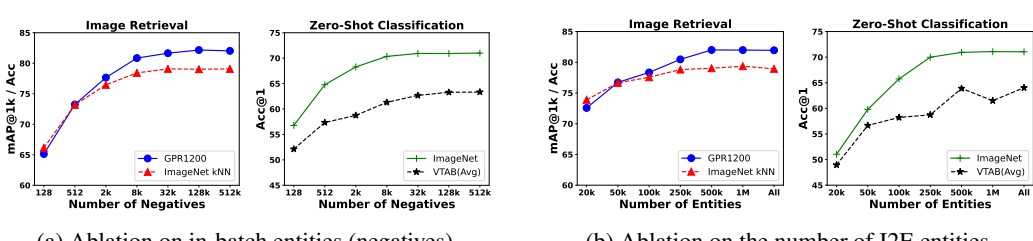

(a) Ablation on in-batch entities (negatives).   (b) Ablation on the number of I2E entities.

Figure 6: **Ablation study** on the number of in-batch negatives and the number of entities to create the I2E dataset.

$N - 1$ effective negatives for each loss calculation. $N = 512k$ is used as the default setting, and Figure 6a shows the ablation of using different number of negative entities during training. Results on image retrieval and zero-shot classification consistently demonstrate a similar pattern, indicating that the continuous addition of negative samples up to 32k leads to substantial improvements in the model's performance. However, introducing additional negatives beyond 32k only brings marginal improvements, lacking significant impact.

**How many entities are needed in I2E?** As indicated in Table 1, the I2E dataset consists of 2 million entities, excluding those with fewer than 5 images. We conduct a more detailed investigation by examining various quantities of entities (Figure 6b). In particular, we select the top number of entities depending on the number of associated images. We start from 20k entities which is similar to the scale of ImageNet 21k, and then select top 50k, 100k, 250k, 1M, and All (2M) entities. Adding more entities consistently improves both image retrieval and zero-shot classification performances until reaching 1M entities. Adding more entities after 1M does not improve the model but also not hurt. We hypothesis the model size may not be large enough to leverage the tail entities, or the current evaluation cannot reveal the potential performance improvement brought up by adding the tail data. Thus, we keep the 2M entities in the dataset, and leave further study for future work.

