# OpenReview forum: "MOFI: Learning Image Representations from Noisy Entity Annotated Images"
_ICLR.cc/2024/Conference — ICLR 2024 poster_

### Official Review · Reviewer_NPAD · 2023-10-27

**Soundness:** 3 good
**Presentation:** 3 good
**Contribution:** 3 good
**Rating:** 6
**Confidence:** 4

**Summary:**

This paper proposes a new vision foundation model, uses three different training strategies, and verifies the performance on the image retrieval task on the constructed dataset. Experimental results on the constructed large-scale data set verify the effectiveness of the proposed new model and training strategy.

**Strengths:**

It looks novel and promising to leverage  Image-to-Entities data to improve image retrieval performance using multi-task pre-training.

Ablation studies are pretty solid and comprehensive to reveal characteristics of the proposed method.

The analysis of different learning strategies is interesting. The paper provides a nice insight into exploiting image representation learning strategies for a particular task.

**Weaknesses:**

In the experiment section, my major concern is that it lacks comparisons with SOTA methods. Ideally, it is encouraged to have comparisons with SOTA methods from both traditional methods and methods based on foundation models. So it will show improvement and advances of the proposed method in this area.

A lack of analyses on efficiencies. In particular, the performance gains from the method should be evaluated along with its time and memory complexity. Does the complex the model, the higher the performance?

The authors observe a similar performance gain on the ImageNet image retrieval task. Why does model performance improve roughly the same on both ImageNet and Image-to-Entities ? The author may want to clarify rationale behind their observations.

**Questions:**

I am wondering why using entity filtering images for multi-task pre-training helps the retrieval problem. This paper could benefit from illustrating more rationale behind using Image-to-Entities dataset.

What are the major difference between multi-task pre-training methods employed in this paper and the contrastive learning?

---

> ### Author Response · Authors · 2023-11-15
> **Response to Reviewer NPAD**
>
> We thank to reviewer for the positive feedback and useful suggestions. Please see responses as follows.
>
> **Baselines**
>
> Thanks for the suggestion, and we will add more baselines as suggested. If you have any particular models you think we should add, please list them and we are happy to include them.
>
> We use CLIP as the major baseline because it is the most commonly vision foundation model now and achieved state-of-the-art performance on various of tasks. It is hard to make fair comparison for other models because of the differences like training data, model size, et.c. For example, CoCa and LiT achieved stronger imageNet zero-shot but with private JFT dataset. Similar to the traditional models.
>
> Besides CLIP, we also considered strong baselines from GPR1200 paper and DINOv2 which is  the state-of-the-art model on image retrieval tasks.
>
> **Performance v.s. Efficiency**
>
> We clarify that the model backbone are the same for MOFI model and baseline CLIP model, the inference latency should be the same. We also keep the seen examples during training for all our models to make the fair comparison.
>
> And we don’t think it is more complex the model, the higher the performance in our case. It is all about the training data, and the right training objective. As reviewer fBZj also commented, *MoFI's classification objective could be seen as pulling similar images together by ensuring they have a similar object distribution*. And many classification tasks and most image retrieval tasks are usually object / entity based, which will benefit more from the entity centroid training objective.
>
> **Performance gain on ImageNet**
>
> We do not fully understand the question here. Could you please elaborate on what you mean by "the performance improves roughly the same on both ImageNet and Image-to-Entities"? Specifically, could you identify the table in question and explain why it poses a problem? We are happy to offer additional clarification once we have a more detailed understanding of your concerns.
>
> **Re Questiones**
>
> We think it is the same as we addressed in the performance v.s. efficiency section above., MoFI's classification objective could be seen as pulling similar images together by ensuring they have a similar object distribution.  As image retrieval tasks are usually object / entity based, they are benefited from the entity centroid training objective. We will add more discussion in the final version of the paper.

---

> > ### Comment · Reviewer_NPAD · 2023-11-17
> >
> > Regarding performance improvements on imagenet, this refers to the penultimate sentence in the second paragraph on the second page of the submission where the author says "We observe a similar performance gain on the ImageNet image retrieval task." I hope the authors provide more analysis of the reasons behind the roughly equal performance improvements on the two datasets.

---

> > > ### Author Response · Authors · 2023-11-17
> > >
> > > Thanks reviewer NPAD for clarifying your question. And sorry for the confusion. When we say a similar performance gain on the ImageNet image retrieval task, we mean the MOFI model also significantly outperforms the CLIP model. We don't mean they have the rough equal performance improvement. The actual numbers could be different, for example:
> > >
> > > | | GPR1200 | ImageNet ImgRet |
> > > |:---| ---: | ---: |
> > > |CLIP_{openai}-L/14| 72.19 | 79.77 |
> > > |MOFI-L/14 | 86.15 | 82.51 |
> > > |Delta | +13.96 | + 2.74 |
> > > We can make the writing more clear in the final version.
> > >
> > > Note GPR1200 and ImageNet image-image retrieval are two tasks in different domains, the number of images to be retrieved are also different. The metric numbers are not really comparable. As ImageNet has been studied for a long time and existing model/dataset could more or less be biased to the dataset, +2.74 points could be as hard as +13.96 improvement on GPR1200.

---

> ### Author Response · Authors · 2023-11-22
> **Looking forward to your reply**
>
> Dear Reviewer NPAD
>
> Thank you again for your insightful reviews of our submission and help to clarify your question after the initial response. Following your feedback, we have provided a detailed response trying to address the concerns you raised. Please let us know if these answered your questions / concerns. We are happy to continue discussing if you have any remaining concerns.
>
> Your effort and time in reviewing our submission are sincerely appreciated.
>
> Warm regards,
>
> Author(s)

---

> > ### Comment · Reviewer_NPAD · 2023-12-05
> >
> > Most of my concerns have been well addressed. Thanks for the responses.

---

### Official Review · Reviewer_4Nss · 2023-10-30

**Soundness:** 3 good
**Presentation:** 3 good
**Contribution:** 2 fair
**Rating:** 5
**Confidence:** 4

**Summary:**

This article primarily presents a billion-scale Image-to-Entities dataset, which includes 1 billion images and 2 million distinct entities. Based on this dataset, the authors have attempted a series of model training algorithms, including supervised learning, contrastive learning, and a combination of both in the form of multi-task learning. The constructed models have achieved state-of-the-art performance on the GPR1200 and VTAB datasets.

**Strengths:**

+ A large-scale image-to-entity dataset is provided, which has the potential to be reused by other pre-training models for vision-language related tasks.
+ Based on this data, a pre-training model has been trained, which can serve as a general-purpose tool for tasks such as image classification and image retrieval.

**Weaknesses:**

+ There are a few unfair comparisons between MOFI and CLIP: (1) CLIP can handle text at the sentence level, serving as the base model for MLLM image inputs. It can be used in tasks like stable diffusion in text-guided image generation. (2) MOFI only utilizes the proposed datasets, which definitely excels at mapping images to entities. The selected dataset for the experiment also leans towards examining the correspondence between images and entities.

+ The training of the MOFI model is limited to image-level correspondence to entities. However, in reality, a single image may contain multiple entities, necessitating the inclusion of region-level correspondence in the model's training. Unfortunately, MOFI does not take this aspect into consideration.

**Questions:**

I am more concerned about how MOFI performs on existing hot tasks, such as whether it is more advantageous to combine with LLM than CLIP, or whether MOFI can be used for image generation tasks.

---

> ### Author Response · Authors · 2023-11-14
> **Response to Reviewer 4Nss**
>
> We appreciate the reviewer for providing valuable feedback and constructive suggestions. While acknowledging the merit of the points raised in the weakness section, we view them more as insightful suggestions rather than inherent weaknesses. Implementing these suggestions has the potential to enhance the MOFI model further. The current results show MOFI advanced the image retrieval SoTA on the GPR1200 dataset by a significant margin, and has strong image representations, with improved zero-shot performance on ImageNet and VTAB benchmarks. We believe it is worth sharing them as the current form, the intuition and results of the paper could benefit the community already.
>
>
> **Regarding the unfair comparison**
>
> Again, we believe it is more like the difference between the two models, not a weakness. And we do not claim that MOFI will replace CLIP in all cases. Table 3 and 4 indeed show the CLIP model and MOFI model can do a better job on different tasks. Users can choose the most suitable model for their use cases.
>
> In addition,  MOFI also has a contrastive task. It is possible to extend the contrastive part to include all CLIP data, which we believe will address the long text understanding concern from the reviewer, but may make the image representation more like CLIP.
>
> **Regarding multiple entities**
>
> Because we extracted the entity from the alt-text, and the nature of short text length of the alt-text, we observed that it usually only describes the primary object / entity in the image. Taking the multi-entity annotation with region (bounding box) is a good direction and we believe it will be the future of MOFI model as well, we leave it as a future work.
>
>
> **Re Questions**
>
> Thanks for the suggestion. We agree combining MOFI with a LLM or an image generation task is interesting and can be useful. However, we believe it is beyond scope of the paper, as it will require a significant amount of space to fully explore the impact when connecting with a LLM or a image generation task. It can be explored as one of the follow-up works.

---

> > ### Author Response · Authors · 2023-11-22
> > **Looking forward to your reply before discussion period ends**
> >
> > Dear Reviewer 4Nss
> >
> > Thank you again for valuable feedback and constructive suggestions. Following your feedback, we have provided a detailed response trying to address the concerns you raised.
> >
> > As the deadline is approaching, would you please let us know if these answered your questions / concerns. We are happy to continue discussing before the discussion period ends if you have any remaining concerns.
> >
> > Your effort and time in reviewing our submission are sincerely appreciated.
> >
> > Warm regards,
> >
> > Author(s)

---

### Official Review · Reviewer_FYZT · 2023-10-31

**Soundness:** 3 good
**Presentation:** 4 excellent
**Contribution:** 3 good
**Rating:** 8
**Confidence:** 4

**Summary:**

This manuscript presents two main contributions. Firstly, it constructs a large-scale I2E dataset in which the correspondences between images and entities are collected. The dataset is derived from a large-scale web corpus, and the authors enhance its data quality by cleaning and filtering. Secondly, the manuscript explores various training recipes and ultimately adopts a multi-task learning strategy that combines supervised learning and contrastive learning. The multi-task model is trained on the constructed I2E dataset to perform image retrieval and classification tasks. Through experiments, the manuscript demonstrates that both the dataset and the training method contribute to achieving better results across multiple downstream tasks.

**Strengths:**

1.	The manuscript constructs a large-scale and high-quality I2E dataset, which can be utilized for model training across various downstream tasks such as image classification and image retrieval. Furthermore, through a comparative analysis between the original CLIP, CLIP trained on I2T, and CLIP trained on I2E, the manuscript demonstrates that the constructed I2E dataset improves model performance.
2.	The experiment is comprehensive, conducting thorough comparisons to assess the impact of different datasets on model performance, thereby validating the high quality of the dataset. Additionally, through comparisons between different models, the manuscript verifies the effectiveness of the proposed training approach.
3.	The paper is well-organized and clearly written.

**Weaknesses:**

1.	The number of training data can be shown in Table 2 for more clear comparison.
2.	Incomplete experimental comparison in Tables 3 and 4. Why is there only CLIP for comparison? Other methods for solving zero-shot/linear probe classification should be discussed and compared.

**Questions:**

Please see Weaknesses.

---

> ### Author Response · Authors · 2023-11-14
> **Response to Reviewer FYZT**
>
> We appreciate the reviewer’s positive feedback and recognition of our contribution. We will add the number of training examples in table 2, and more baselines for table 3 and 4, as suggested.

---

### Official Review · Reviewer_fBZj · 2023-11-01

**Soundness:** 3 good
**Presentation:** 4 excellent
**Contribution:** 2 fair
**Rating:** 6
**Confidence:** 5

**Summary:**

In this paper, the authors propose a vision-language model akin to CLIP, called MOFI. However, unlike CLIP, which primarily focuses on pure contrastive learning with image and text captions, the authors suggest supplementing this by directing the model's attention to specific entities within the captions. They achieve this by integrating an auxiliary task into the standard CLIP training objectives. To accomplish this, the authors introduce a new large-scale dataset, Image-to-Entities (I2E), containing 1 billion images and 2 million distinct entities.

To construct their dataset, the authors initially utilized a large crawled web corpus comprising 8.4 billion image-text pairs. They then applied a named entity recognition model to the text associated with the images, which could be derived from the captions or titles. However, due to the potential for multiple words to refer to the same actual entity, the authors implemented entity linking to connect entities to specific identifiers. To do so, they followed previous methodologies by learning entity embeddings based on graph embeddings from Wikipedia data. To determine the specific identifier linked with the entity, they utilized other entities within the same text to disambiguate. They combined other entity embeddings and iteratively computed the probability of the identifier based on the distance to the full text embedding. To eliminate noisy entities unrelated to the image, the authors used the image-entity distance computed by CLIP to filter out those with low similarity.

The authors explored three strategies for training models on their new dataset. The first strategy involves a straightforward classification-based objective where the aim is to classify an image as one of a fixed number of entities using a standard, fixed-size classifier. The second approach is contrastive pre-training, employing a cross-entropy loss for contrastive training, similar to CLIP. Finally, the authors proposed MOFI, which integrates both a standard contrastive learning objective and the classification-based objective by combining the two losses.

The authors conducted experimental evaluations of their approach across various tasks. Initially, they evaluated their model on the image-to-image retrieval task and exhibited significant gains compared to standard CLIP, controlling for the model backbone architecture. The authors demonstrated that, in many settings, their approach achieved state-of-the-art performance and outperformed other models overall. Next, they evaluated zero-shot classification across ImageNet and VTAB. The authors showcased that their approach achieved state-of-the-art performance on ImageNet and the VTAB Benchmark for their largest model size. Lastly, the authors evaluated the performance of linear probing with their model, illustrating improved performance over CLIP. Additionally, the authors included qualitative results showcasing examples of retrieved images from various benchmarks and a comparison of CLIP and MOFI models on image-to-text and image-to-entity tasks.

**Strengths:**

In terms of strengths, before delving into the technical details of the method, the writing of the paper is very good and appears highly polished. The figures are well-made, and overall the presentation is of very high quality.

In terms of the substance of the paper, the idea of using entities as an additional type of supervision is an interesting one. By explicitly factoring out the entities from the text and then forcing the model to perform a task on those directly, the authors essentially force the model to learn an entity-centric visual representation. In particular, rather than allowing the model to get by with some rough semantic similarity of entities which can be encoded in a feature representation for image-text matching, by explicitly forcing the model to understand millions of specific entities and to disambiguate them, this explicitly forces the model to learn representations for differentiating these very fine-grained entities. Thus, the concept of using entities as an auxiliary task is an interesting one and well motivated.

The authors create a new image-to-entities dataset, which consists of a billion images and 2 million distinct entities. As the authors point out, the dataset has the largest number of distinct class labels than any other dataset, 66 times more than the next one cited. Thus, the dataset created by the authors poses a true challenge, as classifying images into 2 million categories is a daunting task even for the most powerful models. Thus, the image-to-entity dataset could serve as a new benchmark or task for large-scale visual recognition.

In terms of experimental results, the authors demonstrate that MOFI outperforms a number of recent state-of-the-art baselines and a number of image-text foundation models, setting a new state-of-the-art for CLIP-style models on a number of different benchmarks, as they note.

**Weaknesses:**

Despite being an impressive and large-scale model that the authors have trained, in my view, the approach has several significant weaknesses.

Primarily, in terms of technical novelty, there seems to be minimal innovation in the proposed approach. The authors employ a standard contrastive learning approach alongside a standard classification objective, combined through a linear combination. In the context of entity extraction, the authors utilize a named entity recognizer, following an existing approach for learning entity embeddings from Wikipedia. Therefore, in terms of actual technical contribution, the paper seems rather limited. The primary contribution appears to be the extensive scale at which the model operates.

Regarding entity linking, I find the proposed approach weak. It seems the authors don't conduct any form of multimodal entity linking. The process involves initial entity linking using entity embeddings on the text side, followed by a filtering step using a pre-trained CLIP model. A stronger approach for entity linking could potentially be achieved by integrating visual features into the knowledge embeddings. Since this component is central to the proposed approach, such a modification could potentially reduce the number of spurious entity linkings. I'm also curious if the authors experimented with off-the-shelf entity linkers, like BLINK.

Fundamentally, the authors emphasize the significance of the entity prediction task over standard contrastive learning. They argue that the experimental results demonstrate the advantage of predicting these entities. However, this claim is not entirely apparent. For instance, in table 2, the authors report better performance using their approach compared to CLIP and other baselines. Yet, this comparison might be somewhat unfair due to the substantially larger size of the author's dataset. For instance, CLIP's dataset comprises 400 million, whereas the author's dataset is 1 billion. Therefore, a comparison controlling for dataset size would be more appropriate to demonstrate the actual improvement using the author's approach rather than a mere performance increase due to a larger dataset. It would be beneficial to see a comparison with MOFI using 400 million images to understand how it compares to CLIP. Table 3 presents a more precarious situation. For zero-shot classification, we observe that the authors' multitask model doesn't show a clear advantage over CLIP on the ImageNet and VTAB benchmarks. It's not evident that the multitask formulation actually enhances performance compared to CLIP, especially considering the larger dataset size accessible to the authors. The MOFI model slightly outperforms CLIP in linear probing but only marginally.

Additionally, I'd like to ask the authors why they believe the training setting proposed in MOFI is the best approach. Since the advent of CLIP, a considerable amount of work has explored contrastive model training. For example, the LIT paper (Zhai, X., Wang, X., Mustafa, B., Steiner, A., Keysers, D., Kolesnikov, A., & Beyer, L. (2022). Lit: Zero-shot transfer with locked-image text tuning. In Proceedings of the IEEE/CVF Conference on Computer Vision and Pattern Recognition (pp. 18123-18133).) demonstrated significant performance gains by locking the image branch of the model and tuning the text model using a private dataset on which LIT was trained. There are also other large foundation contrastive models like the open-source LAION-H-14 model (https://huggingface.co/laion/CLIP-ViT-H-14-laion2B-s32B-b79K) that the authors could have compared against. These models, not using the described objectives, seem to perform similarly to the author's approach, making this comparison more equitable since the other models the authors compare to seem to have substantially less data.

Regarding the approach itself, it remains unclear if the concept of contrastive learning with a classification objective on entities is necessary. Other works like Coca (Yu, J., Wang, Z., Vasudevan, V., Yeung, L., Seyedhosseini, M., & Wu, Y. (2022). CoCa: Contrastive Captioners are Image-Text Foundation Models.) , demonstrate that a similar objective could be achieved by performing contrastive learning + captioning. For example, on ImageNet, the authors obtained 86.3% zero shot performance using this contrastive captioning-based approach, which significantly outperforms the authors. The contrastive captioning approach actually seems to encompass what the authors are doing in the MOFI paper, as a caption needs to be generated, not just the entity. The authors focus solely on entities, yet other crucial visual relations like events, actions, and visual properties are omitted in their approach. These could be captured under the contrastive captioning setting. Therefore, it's uncertain whether MOFI's approach is superior to training vision-language models as opposed to the contrastive captioning-based approach.

In summary, it's unclear whether the multitask learning approach presented is the optimal way to utilize this data. There's a lack of clear performance analysis against contrastive captioning-based models, making the experimental results challenging to interpret due to the differing dataset sizes.

Additionally, there's a concern regarding the technical approach itself. Filtering out entities using CLIP could potentially result in error amplification. If an entity is poorly captured by CLIP, using it to filter the dataset might exclude that entity.

It seems that the paper would have been better positioned as a very large scale visual entity linking paper, than a vision-language foundation model paper, since the key novelty the authors seem to be addressing is the ability to disambiguate these various entities. However, that itself raises a lot of questions. For example, how do we ensure that more common entities don't overwhelm the long tail of entities? This does not seem to be explicitly handled by the authors' approach and currently authors do not perform an evaluation sufficient to consider the paper for the visual entity linking task.

**Questions:**

1. What is the key benefit of MOFI's technical approach vs contrastive captioners like Coca? It seems that Coca significantly outperforms MoFi on several benchmarks significantly (e.g. Imagenet). Conceptually, the contrastive captioner seems to be more straightforward as well and doesn't require the complex entity linking that MOFI does. It seems that the entity prediction task is a subset of the captioning task that Coca addresses. Also, Coca is capturing entities AND verbs (and other words).

2. Have the authors explored the statistics on how often the model predicts entities from the long tail? Is it mainly focusing on more common entities? It seems the core use of the model would be to classify an image as an entity from Wikipedia - this is an important task for knowledge graph construction and coreferencing, but it is not clear that the authors have evaluated the accuracy on that task - which seems to be the most unique part of the work.

3. Will the dataset and model be released? If not, the contribution would be the technical approach, but as stated above, it is not at all clear that it is significant compared to other ways of training contrastive models (e.g. LiT) on equivalent size datasets AND it is not clear that this multi-task formulation is superior to existing methods for contrastive captioning (Coca).

4. Table 2 - why evaluate on image-image retrieval when CLIP and other baselines were not trained explicitly for image retrieval. These methods have been trained for image-text retrieval, not intramodal retrieval. In contrast, MoFI's classification objective could be seen as pulling similar images together by ensuring they have a similar object distribution. It seems that if we are going to be comparing against VL models like CLIP, LiT, etc. we should focus on those types of tasks (zero shot classification, linear probing, etc.).

---

> ### Author Response · Authors · 2023-11-14
> **Response to Reviewer fBZj [1/2]**
>
> We appreciate the reviewer’s detailed feedback and constructive suggestions. Please find our responses below:
>
> **Contribution**
>
> First of all, we want to clarify that our main contributions are: 1) the way we curate the large scale classification dataset,  2) explore the different training objectives of the constructed dataset,  and 3) demonstrate the effectiveness of the resulting model.
> We advocate for an expansive view of novelty beyond methodology alone. Valuable contributions can stem from innovative experimental designs, the unveiling of new findings or results, or the introduction of novel training paradigms that have the potential to inspire and benefit the community.
>
> Secondly, we agree that both contrastive and classification objectives have been well studied in the research community. However, we want to point out several differences of our work:
>
> 1. Studying learned generic image representation using a noisy large scale classification dataset is new. When combined with the contrastive objective in a multitask setup, we demonstrate that the MOFI model achieves significantly better results on both zero-shot classification and image retrieval. Especially on the image-to-image retrieval benchmark GPR1200, the new model is 14% (absolute) better than OpenAI’s CLIP model.
>
> 2. The existing largest classification datasets people have used are JFT-3B and IG-3.6B, both of them are private and hard to reproduce. Our dataset is curated from public data, can be easily reproduced by others, and 66x larger in terms of number of classes.
>
> 3. We need to adjust the normal classification objective (e.g. sampled negatives and add margin) to effectively learn from such a big number of classes, as detailed in section 3.1.
>
> We will add more discussion about the contributions in the final version.
>
> **Entity Linking**
>
> We appreciate the reviewer’s suggestion on the entity linking, and will be happy to try BLINK as an alternative to our approach. Note that the main purpose of the entity annotation is to provide supervision to learn strong image representation. We find the results from the current noisy annotation have already shown strong performance on multiple benchmarks. Apart from the precision of entity annotation, it's crucial to factor in efficiency, enabling the processing of billions of texts in a feasible time frame. Our current approach can run 8.4B texts within 10 hours. While we think a better entity annotation approach is likely to further improve the performance, we leave it as a future work.
>
> **Dataset size**
>
> Thanks for pointing out the dataset issue. We realized that we didn’t make it clear in the paper. The row 4 (CLIP-B/16_Ours trained on I2T) is an in-house CLIP model trained on the original 5.5B image-to-text data. We train the CLIP model to see the same number of examples to make it a fair comparison. So in table 3 and 4 we also compare the in house CLIP model instead. Hope this addresses your concern. We will clarify it in the paper.
>
> Similarly, table 3 and 4 reports CLIP model trained on several different datasets. It is clear that the CLIP model trained on I2E dataset outperforms the CLIP trained on original image-text pairs. Switching to multitask improves the performance for those entity centroid tasks, while is worse on those tasks that are less entity specific. Users can decide to use which model depends on their tasks. We put more detailed discussion about it in section 4.3.
>
> **Comparing with other CLIP models**
>
> It is hard to compare with other CLIP models as they are trained on different datasets. Some of them are private, e.g. LIT uses JFT-3B. And LAION is prohibited to use by our institution. OpenAI’s CLIP model has been widely adopted so we list it as a baseline model. In addition, we trained our in-house CLIP model using the image-text pairs from the same data source, and a similar number of examples as addressed in the previous paragraph. Given that, we believe our comparison is fair.
>
> We also note that the image representation of LiT model is learned from a classification objective, which is similar to our MOFI_sup-B/16 setting in Table 2, Row 7. MOFI_sup-B/16 has comparable performance on GPR1200, but worse performance on ImageNet retrieval metric.
>
> **Compared to CoCa**
>
> CoCa is trained on private JFT-3B + LAION-1.8B data. It is not a fair comparison to compare our model v.s. original CoCa. We indeed implemented the contrastive captioning model and trained it on our data. However, in our preliminary experiments we didn’t observe the same performance gain compared to CLIP only as it is described in its original paper. We suspect it is because of the private training data (e.g. JFT-3B) they used.
>
> Note, JFT 3B is also a classification dataset with 30k unique labels.

---

> > ### Author Response · Authors · 2023-11-14
> > **Response to Reviewer fBZj [2/2]**
> >
> > We also appreciate the reviewer’s suggestion on positioning the work as a very large scale visual entity linking paper. We are happy to extend the work to study more on the visual entity linking in the future work. However, the motivation of our work is to study the image representation learned from the large scale noisy entity annotated data. The results are strong, we believe it is worth sharing them as the current form and the intuition and results of the paper will benefit the community already
> >
> > **Re Questions**
> >
> > 1. Please see the compared to CoCa section above.
> >
> > 2. Predicting entities is not our primary focus. We can conduct the experiments and report the statistics in the final version. As discussed in the previous paragraphs, we believe there is indeed a lot we can improve in terms of the entity prediction task itself, we leave it as a future work and focus on the learned image representation in this paper.
> >
> > 3. We are working on releasing the pretrained model and dataset. But it will require the approval from our institution, which may take time.
> >
> > 4. Image retrieval tasks are an important category in the community and can power a lot of applications. Embedding based approach is very suitable for this type of tasks.  _Learning semantic similarly from contrastive objective_ is also not new and has been explored before, e.g. [1]. You can think in a similar way that it pushes the image with similar captions distribution together in the embedding space. Our primary goal is to learn generic image embedding without relying on high-quality domain-specific data. We found CLIP is one of the most strong baseline. In fact CLIP doesn’t perform bad at all in these tasks. It performs great, e.g. it outperforms all baseline models on GPR 1200 benchmark from its original paper (also listed in table 2).
> >
> >     In addition, we also compare our model with DINOv2, which is the state-of-the-art model on image retrieval tasks.
> >
> >
> > [1] Y Yang. et, al. 2018, [Learning Semantic Textual Similarity from Conversations](https://aclanthology.org/W18-3022/)

---

> ### Author Response · Authors · 2023-11-16
> **CoCa Experiments**
>
> For your consideration, we also list some preliminary results of CoCa v.s. CLIP in our early exploration. They were trained on the same configurations for fair comparison.
>
>
> |         | ImageNet-zeroshot | GPR1200 |
> | :--- | --------------: | ---------------: |
> | CLIP-B/16 |       68.97         |      61.21 |
> | CoCa-B/16 |    68.08            |  59.93 |
>
> Further, in the CoCa paper, under 4K batch size, CoCa performs much better than CLIP. We indeed also observed this performance improvement. However, as the batch size becomes larger, such as 32k, the performance gain diminishes. We empirically observe that under large batch size setting, CoCa actually performs slightly worse than CLIP.
>
> We did non trivial work to tune CoCa, and also inspected the caption outputs. They all look good and generated captions are very reasonable. So we believe our implementation is correct, but the difference is the training data.
>
> \* *The configurations are different than the models reported in the paper for fast iteration, e.g. using a smaller image-text datasets and were trained with smaller number steps (~300k))*

---

> > ### Author Response · Authors · 2023-11-22
> > **Looking forward to your reply**
> >
> > Dear reviewer fBZj
> >
> > Thank you again for your insightful reviews of our submission. Following your feedback, we have provided a detailed response trying to address the concerns you raised.
> >
> > Given the limited time remaining,  it would be very helpful if you could take a look at our responses and let us know if any ambiguities remain. We would greatly appreciate any further comments regarding our work. And we are more than happy to continue discussing them if you have any remaining concerns.
> >
> > Your effort and time in reviewing our submission are sincerely appreciated.
> >
> > Warm regards,
> > Author(s)

---

> ### Comment · Reviewer_fBZj · 2023-11-22
> **Reply**
>
> Thanks to the authors for the detailed reply comments and apologies for delayed response. I have taken a look at all the reviews and re-read the paper. I also appreciate the authors doing non-trivial experiments to reply to comments.
>
> To be clear, the reason for my weaker initial rating was primarily due to three key factors:
> 1. My perception of the low technical contribution of the paper. In a nutshell, the paper is training like CLIP, with an additional classification objective on top of the vision model. I felt that more advanced types of objectives have been explored before in prior work, e.g. CoCA, Align before Fuse, etc. Seemed like the approach of this paper was quite straightforward - add an object classifier on top of the image branch. It also seemed to go against a line of recent work, e.g. LiT (Locked Image Tuning) that argued that we should in fact *not train* the image encoder, and instead focus on adapting the text encoder. I also felt that the approach could have been executed better, e.g. how to ensure that long-tail objects are predicted? What about some sort of focal loss, etc.? As it stands, my takeaway of the paper was, "they added a classifier to CLIP's image model and got some better/mixed results". Again, LiT and others achieved strong results by locking the image branch.
>
> 2. I felt that some of the claims the paper were making were not experimentally justified for various reasons that I pointed out (e.g. differing dataset sizes that made comparison difficult). In some cases, the results were quite mixed and the story the reviewers were making from the results was not clear-cut from the results. The authors of course point to some successful cases where it outperforms, but others were much more mixed as I pointed out. I think it was equally possible to interpret the results differently, as I mentioned, so I do thinking some of the language making claims about the technique needed to be tempered.
>
> 3. I was concerned about claiming a dataset or model contribution, without a guarantee that the dataset would be released. I also disagree with the authors that "Our dataset is curated from public data, can be **easily** reproduced by others, and 66x larger in terms of number of classes" (emphasis added). I don't think that the dataset can be easily produced by others, or else others would have done something at this scale. As the authors mention in the paper, they use an enormous amount of resources to do the entity linking part from wikipedia. I hope that those models for entity embeddings, etc. that they trained are all released. While of course the authors don't have to release a dataset, we understand that there are various legal and commercial reasons for datasets not to be able to be released, it makes it difficult to then claim this is a major strength of the paper. I think this argument about the dataset would be much stronger if the dataset had some new technique that was used to curate it or process the text, but as the authors mention, the strategy for entity linking largely follows other work on that topic. Thus there isn't really anything novel about the harvesting process or curation process, it is more the scale at which it was done. Even in the comments offered by authors, there is no guarantee that the dataset or models will ever be released. Of course, this may be due to factors beyond authors control, but then it is difficult to place a large emphasis on a dataset contribution from this paper, when the dataset might never be forthcoming.
>
> Regarding new CoCa results:
>
> Thank you very much for providing this. Encourage the authors to include comparable results in main text or supplementary. However, I do have a question.
>
> You state, "Further, in the CoCa paper, under 4K batch size, CoCa performs much better than CLIP. We indeed also observed this performance improvement. However, as the batch size becomes larger, such as 32k, the performance gain diminishes. We empirically observe that under large batch size setting, CoCa actually performs slightly worse than CLIP."
>
> Does this mean you trained CoCa with 4K batch size or the 32K batch size, i.e. are these results from 4K or 32K batched size CoCa? If CoCa works better with a 4K batch size, it would make sense to train it with a 4K batch size, since that is the setting that the original paper claimed worked best. Even though for your CLIP, maybe you use larger batch size, it could be that CLIP does better on larger batch size, but CoCa does not.
>
> Can you clarify this point?
>
> One final comment, I was also looking at papers like OVEN: Open-domain Visual Entity Recognition Towards Recognizing Millions of Wikipedia Entities. In this paper, the authors argue that the same image could reasonably be linked to multiple possible knowledge based IDs, so predict based on a prompt (photo of a horse could be linked to horse, or to the bridle the horse is wearing). So, similar types of image prediction tasks using wiki have been explored.

---

> ### Author Response · Authors · 2023-11-22
> **2nd response**
>
> Thanks reviewer fBZj for taking a look at our responses and follow up question. Really appreciate your time and effort. Please see our latest response below:
>
> 1. As we stated in the original response, we advocate for an expansive view of novelty beyond methodology alone. Valuable contributions can stem from innovative experimental designs, the unveiling of new findings or results, or the introduction of novel training paradigms that have the potential to inspire and benefit the community. To our best knowledge, we believe our findings are novel, and the strong results of training on a large scale entity supervised data could inspire the community for next generation of vision foundation model.
>
> In addition, we also explored the CoCa style captioning training as suggested, by we didn't find the captioning objective helpful in our benchmarks from the preliminary experiments, thus we didn't include it in our main paper. We can include more fair comparison in the final version.
> Regarding LiT, note LiT needs to start from a pre-trained image encoder. Focusing on the text encoder doesn't mean we don't need to train a image encoder. I don't think our results is against it. One possible way is to train a entity supervised classification model first (MOFI_sup) and then train CLIP model with the frozen image encoder. We also did preliminary experiments, and found the results are almost identical. Thus, we kept the multitask training as it doesn't require multi-stage training.
>
> And also note that both of CoCa and LiT are trained using a private JFT dataset from Google, which we cannot access. We believe it could be the reason why we couldn't reach the same level performance as reported by their papers.
>
> 2. We are happy to carefully review the claims again and rewrote to make it clear. We believe our experiments are carefully designed, and make sure all comparisons are fair. In terms of results, we didn't claim people should always use MOFI, e.g. in table 3 and 4, we suggested for tasks are not entity specify, CLIP could be better. We are happy to provide more clarification if reviewer can specify which places are still confusing / concerning.
>
>
> 3. Regarding the dataset contribution. The approach should be robust to the entity embedding model as it is gated by a CLIP model after the entity linking step. e.g. It should be easy to replace it by an open sourced entity embedding model as suggested. We can add an ablation study in the final version.  Once the entity embedding is ready, it only takes 10 hours using 256 machines to extract the entities from the text.
>
> We are also working hard with our institution to release dataset. While cannot promise yet, we are also exploring different ways to make it useful to the community.
>
> One direction we are actively working on reproducing the I2E dataset using open-source datacomp[1] data in lieu of our internal one. By now, we have generated one data version which has similar data scale as the one presented in the paper. It contains 1.19B images and 2.12M entities, which is comparable to the 1.1B images and 2M entities presented in the paper.
>
> With the new I2E data derived from datacomp, we have conducted preliminary model experiments to compare its performance with that of the model trained on internal version. For both training datasets, we maintain the same setup as specified in the paper, except for training the model for shorter time, i.e. 300,000 steps . **We observe comparable performance on GPR1200 MAP@all (a light decrease of -0.7%), ImageNet retrieval Acc@1 (a marginal increase of +0.1%) and ImageNet Zeroshot (a marginal increase of +0.26%).**
>
> We will try our best to release our work to make it useful to the community.
>
>
>
> **Questions for CoCa:**
>
> We experimented training CoCa from 4k batch size to 32k batch size. The numbers we reported above uses 32k batch size for fair comparison.  And  `under 4K batch size, CoCa performs much better than CLIP` is from its original paper. We validated the point in our implementation as well. And here it only means CoCa perform much better than CLIP when they both trained using 4k batch size.
>
>
> At last, thanks for pointing the paper `OVEN: Open-domain Visual Entity Recognition Towards Recognizing Millions of Wikipedia Entities.`  We will take a careful look and comparing them. Note this paper is just published last month at ICCV. We believe it can be treated as a concurrent work.

---

> ### Comment · Reviewer_fBZj · 2023-11-22
> **Reply in Response to Reply**
>
> Thank you for your comments.
>
> I believe that if the authors are intending to replicate their approach on a public dataset (datacomp), have models trained on that, and are planning to release those deliverables, that is a meaningful attempt to contribute something as a dataset, in the event that the original dataset can't be released. Hopefully the models on the datacomp can also be released.
> This will alleviate concerns about data release of the original dataset.
>
> To be clear regarding CoCa result shown, I don't think it is unfair to train each model using the batch size that works best for that particular approach. In fact, I think that it does make it more fair, since original CoCa paper mentioned that 4K batch size works best. I don't think it's incorrect then, to compare CoCa (with 4K batch) against CLIP (with 32K batch), since those are two different models, with different behaviors. In the same way, your method works best with 32K batch, so any comparison should use that, not 4K.
>
> Do you have the number for CoCa (with 4K) vs Your menthod (with 32K)? If so, that is the number that should be reported, presuming that CoCa 4K > CoCa 32K. You seemed to suggest that you did have this number. I am interested in seeing CoCa-B/16 on the Imagenet-zero shot, trained with 4k batch size.
> I ask because your method is less than 1% better than this.
> I am not saying that your paper should be rejected if you don't outperform CoCa with 4K batch size, but for completeness, we should have this to place work in context.
> If accepted, in your final version, I think you should report the version of CoCa that worked best (4K) not the 32k, especially since authors of CoCa mention 4K works better.
>
> If you can share that, that would address all my questions to the authors at this time.
> Thanks

---

> ### Author Response · Authors · 2023-11-22
> **More clarification about CoCa**
>
> We are sincerely grateful to the reviewer fBZj for engaging in a thoughtful and active discussion!
>
> Regarding CoCa, we apologies for any confusion in our communication about it.  We experimented for both CoCa and CLIP with different training batch size from 4k to 32k.  Under 4K batch size, CoCa performs much better than CLIP relatively (in our experiments, we saw +2 points (+8% relative) improvement on ImageNet zero-shot). But 32k batch size trained CoCa still perform better than the CoCa model trained under 4K batch size. See the table below. _Note they are trained using the same number of training steps, so technically the 32K models have seen 8x more training examples than the 4k models._
>
> |         | Batch size | ImageNet-zeroshot |
> | :--- | ---: |--------------: |
> | CLIP-B/16  | 4k | 33.69 |
> | CoCa-B/16 | 4k | 35.67 |
> | CLIP-B/16  | 32k |      68.97         |
> | CoCa-B/16 |  32k |   68.08            |
>
>
> We emphasize that despite putting considerable effort into training the captioning objective, we have not observed any improvements upon its addition.  It is often challenging to claim something does not work, because there are so many things to explore, including those we do not have control, e.g. JFT dataset. However, based on our experiments, the inclusion of the captioning objective has not proven beneficial so far.
>
> \*_We also note in CoCa paper, the final model is trained using 65k batch size. In our practice, we found models trained with >=16k batch size can get very similar performance as long as they have seen the same number of the training examples. Larger batch size with more resources can make the model interate faster._

---

> > ### Comment · Reviewer_fBZj · 2023-11-22
> > **Thank you!**
> >
> > Thanks so much. This has addressed my questions.

---

> > > ### Author Response · Authors · 2023-11-23
> > > **Thank you!**
> > >
> > > That's fantastic. Thank you for the insightful discussion. We would greatly appreciate it if you could reconsider the score if you now have a more positive view of our paper.

---

### Author Response · Authors · 2023-11-23
**Response summary**

Dear reviewers and ACs

We would like to express our sincere gratitude for the invaluable feedback provided on our paper. Your constructive suggestions and positive acknowledgments are instrumental in refining our work.

We summarize our contributions as follows:
 1) A simply yet effective approach to curate the large scale classification dataset
 2) Exploration of diverse training objectives on this constructed dataset
 3) Demonstration of the effectiveness of the resulting model.

We feel very encouraged by the unanimous recognition from all reviewers regarding the significance of our dataset and the scalable approach we used to construct the dataset. Notably, Reviewers fBZj and NPAD highlighted the novelty of using entities as supervision for learning image representation—an aspect we believe enhances the uniqueness of our work.  Reviewers FYZT and NPAD also commented the solidity and comprehensiveness of our experimental design, affirming the strength and convincing of our results.

A special appreciation goes to Reviewer fBZj for a thorough / detailed review and actively participating the discussion. Reviewer fBZj recognize several positive aspect of the paper, and the major concern is the technical novelty of the model employed in the paper.  We advocate for an expansive view of novelty beyond methodology alone. Valuable contributions can stem from innovative experimental designs, the unveiling of new findings or results, or the introduction of novel training paradigms that have the potential to inspire and benefit the community. To our beset knowledge, we are the first to study the image representation learned from the large scale noisy entity annotated data. The robustness of our results reinforces our belief in the relevance of sharing this work, with potential benefits for the wider community.  We are also trying our best to open source the dataset and pre-trained model to the community, upon our institution’s approval. During the insightful discussion, reviewer fBZj also raised additional valuable points, such as the paper's positioning, potential extensions, and the inclusion of an additional CoCa (contrastive captioner) baseline. We have comprehensively tackled these inquiries, providing elaborations and presenting additional experimental results. For a detailed account of our responses to these concerns, we refer you to the specific feedback section.

Furthermore, Reviewer 4Nss offered insightful suggestions in the weaknesses section, particularly regarding a more in-depth discussion of the distinctions between CLIP and MOFI, as well as the potential extension of incorporating multi-entity with region annotation.  We acknowledge these as excellent points. These points, however,  may not be treated as weaknesses of the paper but more like a question and a future work.

In our responses to each reviewer, we have meticulously addressed all comments and questions raised. We believe that these discussions have significantly enhance our paper, and we look forward to further discussion in enhancing the quality and impact of our work.

Thank everyone again for your time, expertise, and thoughtful engagement with our research.

---

### Meta-Review · Area_Chair_wwH9 · 2023-12-06

**Metareview:**

Many questions were raised and discussed during the rebuttal phase. Most reviews remain borderline (5's and 6's) after the rebuttal phase, but several reviewers explicitly comment that most of their questions have been resolved.

**Justification For Why Not Higher Score:**

only one 8 but from a brief review

**Justification For Why Not Lower Score:**

only one 5

---

### Decision · Program_Chairs · 2024-01-16

Accept (poster)